# IgG antibodies to SARS-CoV-2 in asymptomatic blood donors at two time points in Karachi

**Muhammad Hasan[1], Bushra Moiz[1], Shama Qaiser[1]☯, Kiran Iqbal Masood[1]☯, Zara Ghous[1], Areeba Hussain[1], Natasha Ali[1], J. Pedro Simas[2], Marc Veldhoen[2], Paula Alves[3], Syed Hani Abidi[4], Kulsoom Ghias[4], Erum Khan[1], Zahra Hasan[1]* **

1 Department of Pathology and Laboratory Medicine, Aga Khan University, Karachi, Pakistan, 2 Instituto de Medicina Molecular, Universidade de Lisboa, Lisbon, Portugal, 3 IBET ITQB, Universidade NOVA de Lisboa, Lisbon, Portugal, 4 Department of Biological and Biomedical Sciences, Aga Khan University, Karachi, Pakistan

☯ These authors contributed equally to this work.
* zahra.hasan@aku.edu

**Data Availability Statement:** All relevant data are within the paper and its Supporting Information files.

**Funding:** This work was supported by the Provost's Academic Priorities Fund, The Aga Khan

## Abstract

### Introduction

An estimated 1.5 million cases were reported in Pakistan until 23 March, 2022. However, SARS-CoV-2 PCR testing capacity has been limited and the incidence of COVID-19 infections is unknown. Volunteer healthy blood donors can be a control population for assessment of SARS-CoV-2 exposure in the population. We determined COVID-19 seroprevalence during the second pandemic wave in Karachi in donors without known infections or symptoms in 4 weeks prior to enrollment.

### Materials and methods

We enrolled 558 healthy blood donors at the Aga Khan University Hospital between December 2020 and February 2021. ABO blood groups were determined. Serum IgG reactivity were measured to spike and receptor binding domain (RBD) proteins.

### Results

Study subjects were predominantly males (99.1%) with a mean age of 29.0±7.4 years. Blood groups were represented by; B (35.8%), O (33.3%), A (23.8%) and AB (7%). Positive IgG responses to spike were detected in 53.4% (95% CI, 49.3–37.5) of blood donors. Positive IgG antibodies to RBD were present in 16.7% (95% CI; 13.6–19.8) of individuals. No significant difference was found between the frequency of IgG antibodies to spike or RBD across age groups. Frequencies of IgG to Spike and RBD antibodies between December 2020 and February 2021 were found to be similar. Seropositivity to either antigen between individuals of different blood groups did not differ. Notably, 31.2% of individuals with IgG antibodies to spike also had IgG antibodies to RBD. Amongst donors who had previously confirmed COVID-19 and were seropositive to spike, 40% had IgG to RBD.

University. The funders had no role in study design, data collection and analysis, decision to publish, or preparation of the manuscript.

**Competing interests:** The authors have declared that no competing interests exist.

## Conclusions

Our study provides insights into the seroprevalence of antibodies to COVID-19 in a healthy cohort in Karachi. The differential dynamics of IgG to spike and RBD likely represent both exposure to SARS-CoV-2 and associate with protective immunity in the population.

## Introduction

COVID-19 is a highly contagious respiratory infection caused by SARS-CoV-2 [1, 2]. COVID-19 was declared a pandemic in March 2020 by the World Health Organization. The first case of COVID-19 in Pakistan was diagnosed on February 26, 2020. An estimated 1.5 million cases have been reported in Pakistan with death toll of 30,333 with a case fatality rate (CFR) of 2% (last accessed 23 March 2022) [3]. The first wave of the pandemic in Pakistan peaked in June 2020, when approximately 5000–6000 cases were detected each day [3]. The second wave occurred between November 2020 and January 2021, the third between March and May 2021 and the fourth wave [4]. Overall, the magnitude of new daily COVID-19 cases identified has been lower than in many parts of the world such as for example, the USA with 79.8 million cases at an incidence of 6154 per 100,000 population, CFR 1.2% (last accessed 23 March 2022) [5]. The number of COVID-19 cases reported is impacted by access to testing which has varied between countries. Compare that in the USA greater than 10,000 tests are conducted per 1 million individuals based on need [5]. In January 2021, 40,000 SARS-CoV-2 PCR tests or approximately 200 tests per one million population were conducted in Pakistan [6].

The majority of individuals with COVID-19 tend to be asymptomatic or with minimal symptoms [7] and are unlikely to be diagnosed and reported after laboratory based confirmation [8]. Therefore, it is important to have alternate strategies to identify infections and understand COVID-19 transmission to inform strategies for prevention and control of SARS-CoV-2 infections. Seroprevalence assays can help estimate disease burden in the population. A countrywide study of seven of the most populous cities in Pakistan conducted in July 2020 showed seroprevalence to vary between 31.1 and 48.1% [9]. Zaidi et al. showed an average COVID-19 related seropositivity of 36% between April through July 2020 in Karachi, varying between industrial employees (50%), community (34%) and healthcare workers (13%) [10].

The above mentioned studies measured antibodies to nucleocapsid (N) protein of SARS-CoV-2 and it has been shown that seroprevalence estimates may vary based on the assay used [11]. The spike protein of SARS-CoV-2 is highly immunogenic [12] and antibodies to spike protein are associated with protective immunity against the virus [13]. Here we measured IgG to both spike and RBD protein to investigate immunity against SARS-CoV-2 in a population of voluntary blood donor who represent healthy individuals in the community. We focused on donors at a tertiary care facility in Karachi during the second COVID-19 wave from December 2020 until February 2021.

## Materials and methods

### Study setting

The study was conducted at Aga Khan University Hospital (AKUH) from December 2020 to February 2021. AKUH is a 700-bedded hospital with facilities of trauma, surgery, and bone marrow transplants. The blood bank at AKUH has an estimated annual collection of 25,000 to 30,000 units of whole blood from healthy non-remunerated blood donors. Approximately 90%

of these blood donors are exchange donors donating blood for admitted patients. The AKUH Clinical Laboratories are accredited by the College of American Pathologists and follow national and international standards for blood collection, manufacturing, storage, and transportation. Each blood donor is registered for donation following an interview with written history taking to exclude the presence of transfusion transmitted infections and to ensure donor safety. Since the COVID-19 pandemic, questions related to the risk of having COVID-19 infection in previous 4 weeks was added in the donor screening. Any donor with a history of COVID-19 related symptoms or confirmed infection was excluded from enrollment as a blood donor. However, this exclusion criteria did not prevent recruitment of those who were asymptomatically or paucisymptomatically infected at the time of donation.

Each donated blood unit was typed on automated gel platform (IH-1000, DiaMed GmbH, Cressier FR, Switzerland) and screened for malaria (ICT, BinaxNOW, Abbott), syphilis (latex agglutination, Sfilide RPR, Milano, Italy), viral hepatitis B, C and HIV (Chemiluminescence assay, Vitros ECiQ Immunodiagnostic system, Orthoclinical diagnostics, Johnson & Johnson, United States).

## Participants and research strategy

The study was approved by Ethical Review Committee of Aga Khan University (study #2020-5152-11688) and was conducted according to good clinical practices and the Declaration of Helsinki. Adult blood donors aged 18 years and over who presented to the blood bank for donations between December 2020 and February 2021 were informed about the study. Participants were recruited with written informed written consent.

Serum was collected and tested for the presence of IgG antibodies to spike. Once IgG antibody responses were available, those with a positive result were contacted by phone for further information including, a risk assessment of COVID-19 infection through secondary questionnaire. This included information regarding any prior history (within the six months) of respiratory illness/flu-like symptoms, domestic/international travel history, or contact with individuals who were suspected or confirmed for COVID-19 PCR or prior SARS-CoV-2 antibody testing. A 'case' of COVID-19 was identified as those with either a positive SARS-CoV-2 PCR or positive IgG antibodies using any commercially available laboratory diagnostic test. Exposure to a suspected or confirmed case of COVID-19 was self-reported by the blood donors, their laboratory reports were not available for review and confirmation.

## Laboratory analysis

**Sample collection.**   From each donor, 4 ml of blood sample was collected in gold-top serum separator tube (BD vacutainer® blood collection tube. The serum was separated, and aliquots were stored at -80˚C till further analysis.

**ELISA for IgG to spike and RBD.**   Recombinant spike and RBD protein were obtained from IBET ITQB, NOVA University, Portugal. All serum samples were tested in duplicate using in-house enzyme linked absorbent assay (ELISA) for SARS-CoV-2 antibodies as described by Stadlbauer et al. [14] and detailed by Figueiredo-Campos et al. [15].

For ELISA against SARS-CoV-2 spike and/or RBD protein, a 96-well ELISA plate was coated with 50μl recombinant antigen at a concentration of 2 μg/ml in PBS. Briefly, the wells were blocked with 200 μl of PBS + 0.1% Tween (PBS-T) + 3% non-fat milk. After washing, 100 μl each of serum samples diluted 1:100 in PBS-T + 1% non-fat milk powder were added to the plate and incubated for 2 hours at room temperature. Wells were stained with goat anti-human IgG Fc (HRP). The plate was developed using TMB substrate solution, stopped with 0.5M sulfuric acid and optical density read at 450nm.

For assay validation, sera from 45 COVID-19 convalescent cases, drawn 4 weeks after their PCR confirmed diagnosis, were used as positive controls. Sera from 55 healthy individuals from the pre-pandemic period were used as negative controls. IgG antibody results for the 100 control individuals are depicted in Fig 1A. These data were used to calculate the sensitivity and specificity of the ELISA assays using 0.5 OD450 nm as a cut-off for positive results for both assays. The sensitivity of the ELISA for IgG to spike was found to be 100% (92.1–100, 95% CI) with a specificity of 100% (93.5–100, 95% CI). The area under the ROC curve for IgG to spike was 0.952, p value < 0.0001, Fig 1B. The sensitivity of the ELISA for IgG to RBD was found to be 91.1% (78.8–97.5, 95% CI) with a specificity of 94.6% (82.4–98, 95% CI). The area under the ROC curve for IgG to RBD was 0.913, p<0.0001, Fig 1C.

Pooled positive and negative control sera were used on each plate, with positive samples run as a standard curve comprising a titration of sera (S1 and S2 Figs). Serum samples of study subjects were tested for IgG antibodies to spike and those with a reactive result were further tested for IgG to RBD. All samples negative for IgG to spike were positive for IgG to RBD.

### Statistical analysis

Data (S1 Appendix) was analyzed through SPSS version 24. Normality of data was checked through Shapiro walk test and mean ±SD was used for normally distributed and median (IQR) for skewed continuous data. Frequency or proportion was used to give estimates for categorical data. Chi-square test was used to compare the frequencies of various antibodies with respect to age groups (with 30 years as the cutoff) and blood types and the threshold of significance was a *p-value* <0.05.

## Results

### Demographics of the study population

The AKUH blood bank approached 1809 healthy non-remunerated blood donors regarding this study between December 2020 and February 2021. Five hundred and fifty-eight subjects

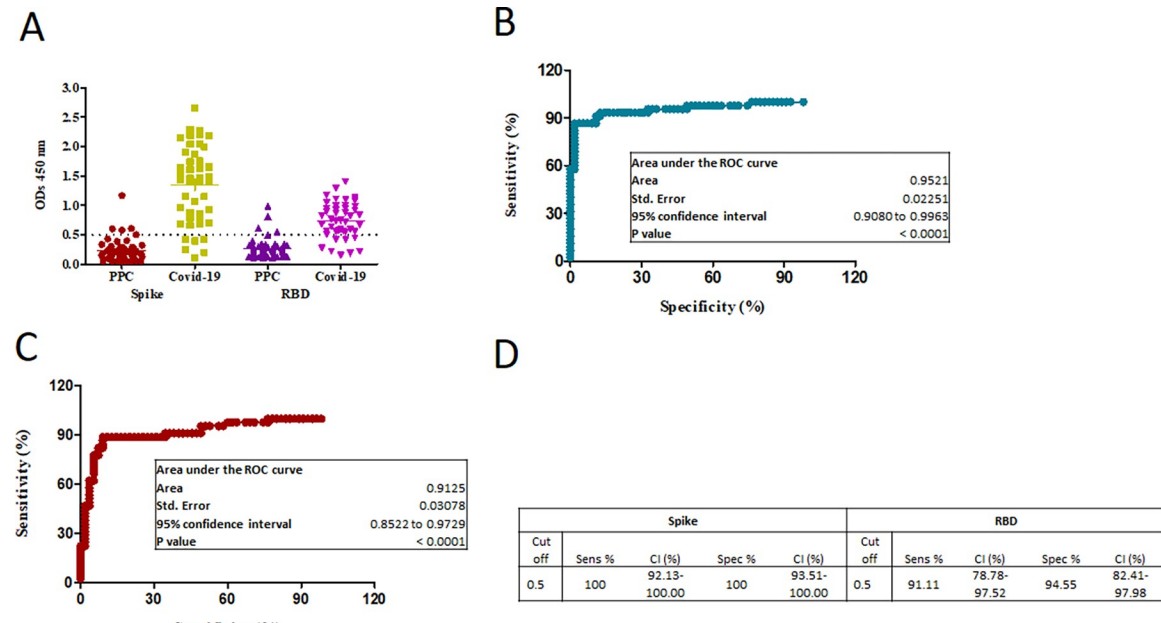

**Fig 1. IgG antibody responses to spike and RBD in COVID-19 cases and pre-pandemic controls (PPC).** The graphs depict A, IgG levels to Spike and RBD in COVID-19 cases (n = 45) and PPC (n = 55). B, shows the sensitivity and specificity calculation for IgG to spike and RBD using an ELISA cut-off of 0.5 at 450 nm. C, ROC curve of IgG to spike and D, RBD for the positive and negative cases.

(31%) consented to participate in the study and submitted blood samples for the purpose. Samples were collected in two sets, the first set of 322 samples were collected in December 2020 and the second set of 236 samples was collected in February 2021.

Study subjects included 553 (99.1%) males and 5 (0.9%) females with a mean (±SD) age of 29.0±7.4 years (range 17–53 years). The individuals across age groups were found to be 40.1% (17–25 y), 41.9% (26–35 y), 15.2% (36–45 y) and 2.7% (46–55 y), respectively, Table 1. Therefore, there was a significantly greater number of individuals younger individuals aged 35 years (82.1%).

The study subjects stratified by their ABO blood groups were found in decreasing order to represent B (35.8%), O (33.3%), A (23.8%) and AB (7.0%) blood groups, S1 Table. Of note, there was no difference found between the number of individuals with A, B, AB or O blood groups when compared across age groups of study subjects.

## IgG to spike and RBD proteins in study subjects

Donor sera were tested for the presence of IgG antibodies to spike of SARS-CoV-2. Of the 558 individuals tested, IgG to spike protein was detected in 298 (53.4%) blood donors (Fig 2A). Ninety-three study subjects had IgG antibodies to RBD, comprising 16.7% of all blood donors. Thirty-one per cent of individuals seropositive for IgG to spike also had IgG antibodies to RBD. All study subjects tested and found negative for IgG to spike were also negative for IgG to RBD. A correlation analysis between IgG titers to spike and RBD revealed a moderately significant positive correlation, (p value < 0.0001, r = 0.3787) between the two data sets, Fig 2B.

## Prior symptoms, risk factors and history of COVID-19

Of 298 study subjects who had positive IgG antibodies to spike, only 190 (63.7%) could be contacted for additional information. Of these 37.4% had a history of respiratory illness, 24.7% had a history of travel outside their home city, 17.9% had contact with suspected patient and 14.7% had contact with a PCR positive COVID-19 case, Table 2.

During donor interview, 56/190 (29%) individuals reported that they had undertaken a diagnostic SARS-CoV-2 laboratory test through either, a PCR on a respiratory sample (n = 44) or blood based antibody testing (n = 12). Eleven individuals (25%) of individuals who underwent PCR testing had a positive PCR result confirming SARS-CoV-2 infection. Whilst all 11 were Spike IgG positive, only four (36.4%) had IgG antibodies to RBD.

Of the 12 blood donors who had undergone COVID-19 antibody testing earlier, four (33.3%) had received a positive antibody test result earlier. Of these, we found only two (50%) individuals to have positive IgG antibodies to RBD. Therefore overall, only six (40%) with previously confirmed COVID-19 had positive IgG antibodies to RBD.

**Table 1. Frequency of IgG to spike and RBD in various age bands of the study group.**

| Age groups (years) | n (% individuals of total group) | individuals with IgG to spike (% of total) | age-adjusted % of individuals with IgG to spike | IgG to spike (95% CI) | individuals with IgG to RBD (% of total) | age-adjusted % of individuals with IgG to RBD | IgG to RBD (95% CI) | % population prevalence* |
|---|---|---|---|---|---|---|---|---|
| 17–25 | 224 (40.1) | 49.5 | 39.7 | 33.3–46.1 | 14.2 | 36.5 | 30.1–42.8 | 21 |
| 26–35 | 234 (41.9) | 56.4 | 29.6 | 23.8–35.5 | 20.1 | 33.8 | 27.8–39.9 | 14.9 |
| 36–45 | 85 (15.2) | 57.6 | 20.9 | 12.2–29.5 | 15.3 | 17.8 | 9.6–25.9 | 10.3 |
| 46–55 | 15 (2.7) | 40.0 | 9.7 | -5-24.7 | 6.6 | 5.1 | -6-16.3 | 6.9 |
| *p-value* | | | *0.297* | | | *0.730* | | |

Chi-square test of association was applied and considered significance level at α<0.05.

* as per population census of Pakistan (2017), Pakistan Bureau of Statistics

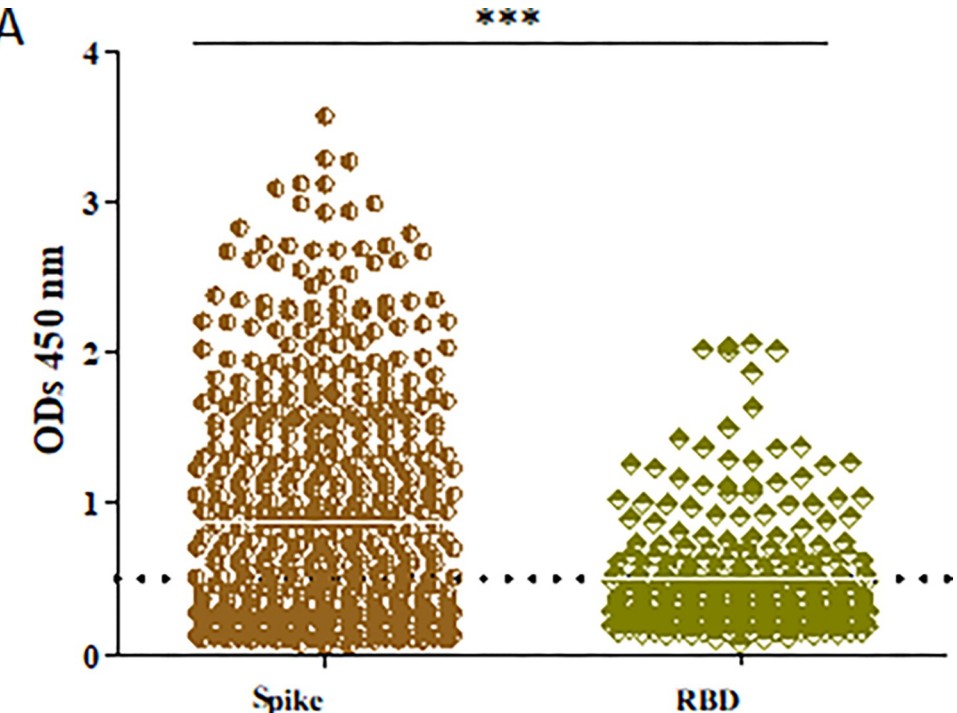

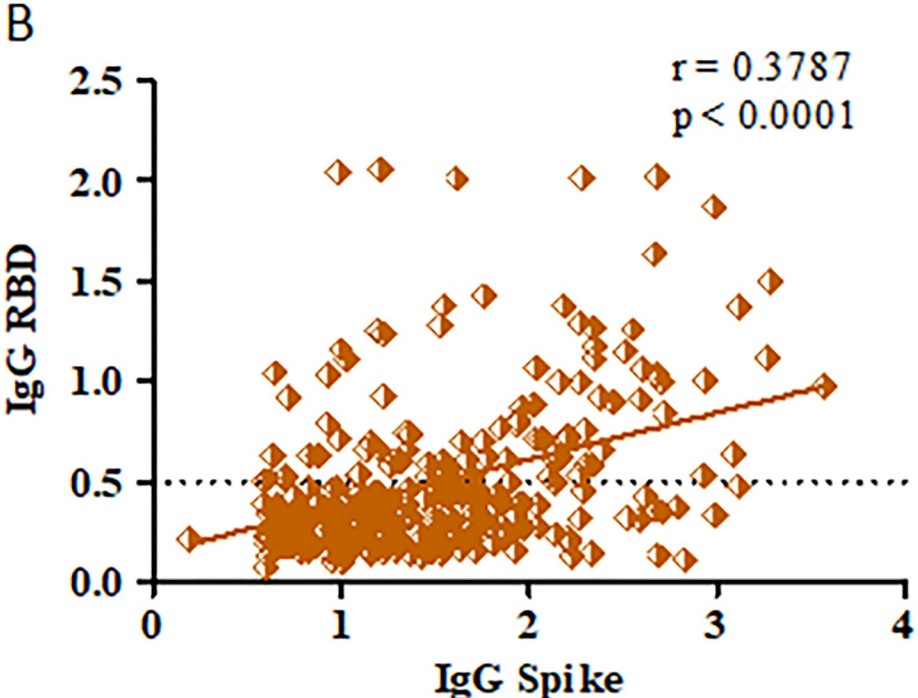

**Fig 2. IgG antibody in 558 healthy blood donors.** A, IgG positivity was determined in sera against Spike and RBD protein. IgG to Spike was measured in 558 individuals. IgG to RBD was determined in the 298 blood donors who had a positive IgG to Spike. B, A depicting results for IgG to spike/RBD, show p value $< 0.0001$ and $r^2 = 0.0678$. Spearman's rank correlation was run between IgG to Spike and RBD. r = 0.3787, 95% confidence interval: 0.2701 to 0.4777 and P value (two-tailed) $< 0.0001$ indicate a positive correlation between the two data sets. The dotted line in 'A and 'B' represents the cut-off for a positive IgG result in each case.

**Table 2. Clinical characteristics of blood donors who had a positive IgG to Spike protein.**

| Clinical details of donors | Spike IgG N/total donors (%) | RBD IgG N/spike IgG (%) |
|---|---|---|
| Flu- like symptoms | 71 (37.3) | 22 (30.9) |
| History of overseas travel | 47 (24.7) | 10 (21.3) |
| Contact with suspected COVID-19 case | 34 (17.9) | 9 (26.5) |
| Contact with confirmed COVID-19 case | 28 (14.7) | 6 (21.4) |
| Prior history of COVID-19 (PCR confirmed) | 11 (5.8) | 4 (36.4) |
| Prior history of COVID-19 (antibody confirmed) | 4 (2.1) | 2 (50) |

One hundred and ninety individuals with a positive IgG to Spike could be contacted for information provided. The information depicts the number of individuals who had an affirmative result for each category.

## Demographics of individuals with a positive IgG response to spike and RBD

The mean age of individuals who had a positive IgG titer to spike was 29.2±7.2 years. Also, the male predominance of study subjects who were IgG positive to spike (n = 297; 99.7%) was comparable to that of the overall cohort.

The frequency of individuals with positive IgG antibodies to spike and RBD was compared across age groups; 17–25, 26–35, 36–45 and 46–55 years. There was no difference across age groups between individuals with sera reactive to spike (mean 53.4%; CI 95%, 49.3–37.5, *p-value* 0.274), Table 1. Similarly, the proportion of individuals with positive IgG to RBD were also comparable across age groups studied (mean 16.7%; (95% CI, 13.6–19.8%, *p-value* 0.462).

## Blood groups amongst study subjects with IgG to spike

Blood donors with blood groups A, B, AB, and O who had positive IgG antibodies to spike are shown in Fig 3. These were found to be 52.6%, 56.5%, 43.5% and 52.6% across the blood groups respectively, S1 Table. Sera tested for IgG to RBD showed antibody positivity of 37.1%, 32.7%, 17.6% and 27.6% across A, B, AB and O blood groups in that order (Fig 3, S1 Table). There was no statistically significant difference in the frequency of positive IgG responses to

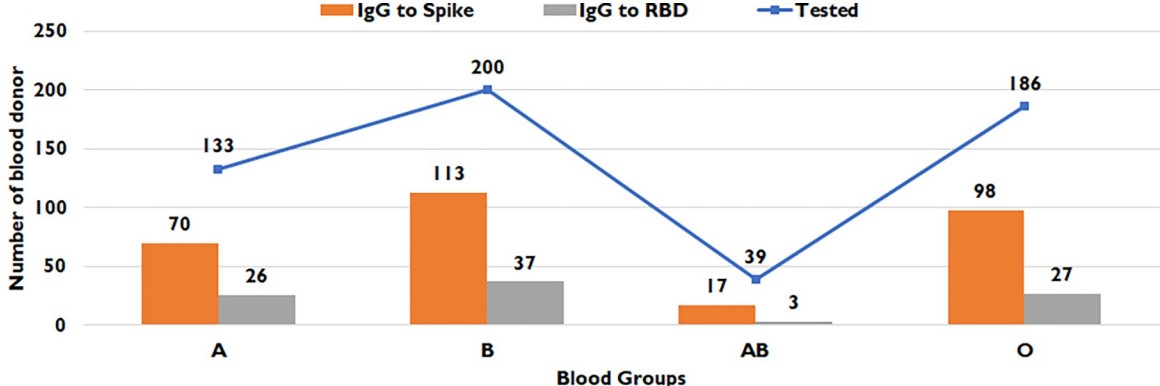

**Fig 3. Seropositivity in blood donors in relation to their ABO blood groups.** The graphs show the number of individuals tested (blue bars), those who had a positive IgG to spike (orange bar) and those who had a positive IgG to RBD (grey line) in each case. Data for individuals is shown as per their A, B, AB and O blood groups.

**Table 3. Comparison of IgG antibodies to SARS-CoV-2 spike and RBD proteins in December 2020 and February 2021.**

| | Dec-20 | | | Feb-21 | | | |
|---|---|---|---|---|---|---|---|
| | n | % IgG positive | 95% CI | n | % IgG positive | 95% CI | p-value |
| Spike | 174 | 54.2% | (48.8–69.7) | 124 | 52.3% | (46.0–58.7) | 0.727 |
| RBD | 174 | 34.5% | (27.4–41.5) | 124 | 26.6% | (18.8–34.4) | 0.148 |

Chi-square test of association was applied and considered significance level at α<0.05.

either spike (*p-value* 0.503) or RBD (*p-value* 0.342) antigens across the different age groups studied.

## Month-wise seroprevalence of SARS-CoV-2

We compared the frequency of COVID-19 antibody positivity between December 2020, after the start of the second wave, and in February 2021, at the end of the wave (https://ourworldindata.org/coronavirus/country/pakistan). No statistically significant difference (p = 0.805) was observed in the frequency of positive IgG responses to spike in blood donors sampled between December 2020 (mean 54%, 95% CI; 48.8–69.7) and February 2021 (mean 52.5%, 95% CI; 46.0–58.7). Similarly, there was no difference between the proportion of individuals with positive IgG to RBD in December 2020 (mean 34.4%; 95% CI; 27.4–41.5) and February 2021 (mean 26.6%, 95% CI; 18.8–34.4), Table 3. Further, the frequency of individuals with positive IgG antibodies to Spike and RBD between the months of December 2020 and February 2021 were compared as per blood groups.

There was no difference found between IgG to spike or RBD positivity between individuals of A, B, AB or O blood groups tested in December 2020 or February 2021, S3 Fig.

## Discussion

Sero-epidemiological studies are helpful in identification of the magnitude of disease in a population by estimating the number of individuals with subclinical /asymptomatic infections. Further, they can provide insights into the immune protection present in the population. In this study of healthy blood donors tested at the time of the second COVID-19 wave in Pakistan between December 2020 and February 2021, we found 53% to have positive IgG antibodies against the spike protein of SARS- CoV-2. This seroprevalence is significantly higher than the previously reported data for healthy blood donors from Europe such as; 0.9% in Italy, 1.9% in Denmark, 2.7% in France and 0.91% in Germany [16–18] [19]. A study conducted in healthy blood donors in Lombardy, Italy [8], one of the first lock down region in Italy reported a frequency of 19.7% for SARS-CoV2 anti-S1 and anti-S2 IgG antibodies and 21.6% for neutralizing antibodies during March to June 2020 [20]. SARS-CoV-2 seroprevalence of 4.0% was observed in 2857 blood donors from Brazil in April 2020 [21] while New York, United States observed a seroconversion of 1 in every 8 blood donors in June-July 2020 [22]. A high level of exposure to the virus was observed in Saudi Arabia, where IgG to spike proteins was observed in 19.3% during May to July 2020 [23]. Moreover, an increase in seroprevalence was reported in blood donors in parallel with the rise of COVID-19 infection in Jordan [24]. Antibodies to SARS-CoV-2 antibodies are shown to persist for up to 6 months post-disease onset [15]. With seroprevalence rising over the time period tested during the pandemic period of 2020 and 2021 [8].

Reports of healthy blood donors from Karachi tested in June 2020 (at the peak of the first COVID-19 wave) showed a seropositivity in 15/70 (21.4%) with an increase (to 37.7%) in July

2020 [25]. Seroprevalence data from Karachi has showed COVID-19 antibody positivity to rise to 21.8% in high transmission neighbourhoods in Karachi by August 2020 [26]. The current study performed December 2020 until February 2021 at the time of the second COVID-19 wave shows a further rise (to 53%) in seropositivity in Karachi. On January 1, 2021, 41,000 tests were conducted each day and approximately 2500 positive COVID-19 cases (6.1%) were confirmed within a 24 h period [6]. Our data therefore alludes to much greater infection rates in Karachi than identified from PCR based testing alone. A higher seropositivity is suggestive of active disease transmission in the community and this may be related to over population, congested living conditions and low compliance with the SOPs like wearing mask in public and maintaining social distancing.

Our data showed a comparable frequency of seropositivity to spike and RBD across the age groups studied. Our study group was relatively young, mainly ranging from their second to fourth decades of life. Notably, we had very few individuals aged 50 years and over as it was younger individuals who comprised the blood donor cohort.

Though several population-based studies have demonstrated a high seroprevalence in males compared to females [18, 21, 27, 28], such a comparison could not be undertaken because of gender imbalance in the current study with predominantly more male donors. Another limitation of this study is that a detailed COVID-19 history was not taken prior to enrollment of study subjects and individuals who did not report any signs or symptoms of disease within the past four weeks was eligible for the study. We were therefore unable to rule out those who may have had asymptomatic or subclinical infections at the time of recruitment.

Previous studies have shown COVID-19 to be present more in individuals with blood group A [23]. as compared to those having blood group O [29]. However, here, we did not find any significant difference in seroprevalence among different ABO blood group types, corroborating an earlier report from Pakistan [25].

One or more symptoms associated with COVID-19 including fever, cough, and loss of smell/taste, fatigue, and malaise were observed in 71 (37.4%) individuals. It may be that up to 60% of the individuals who were infected by SARS-CoV-2 remained asymptomatic. This finding is congruent with a study in Quebec where half to two third seropositive donors were found to be asymptomatic [30]. Of those with IgG antibodies to spike who had undergone diagnostic PCR testing or antibody testing for SARS-CoV-2, only 11 (25%) were PCR positive. The reasons for this low PCR positivity are not known, but a low viral load at the time of PCR testing, inadequate sampling technique or variability in PCR-based diagnosis may be the possible explanations. Of the spike seropositive blood donors who had undergone COVID-19 antibody testing earlier, 4 (33.3%) had a prior positive test result. Importantly, of these 15 individuals with prior laboratory confirmed COVID-19, only 6 (40%) had IgG antibodies to RBD. These data highlight the dynamic the nature of antibody responses to SARS-CoV-2 and also the variability of test results depending on the assay target employed.

Interestingly, in our study we observed a discrepancy between individuals who had positive IgG to spike and those who were positive to RBD. Overall, only 31.2% subjects with IgG to spike demonstrated IgG to RBD. This ratio of spike to RBD positivity was comparable across the age groups investigated.

We found a significant correlation between IgG to spike and RBD, correlating with previous reports [15]. Further, our RBD ELISA was 94.6% specific (95% CI; 82–98) and 91% sensitive (95% CI; 79–98) for diagnosis of COVID-19 cases. Previously, it has been shown that IgG to spike and RBD both demonstrate high specificity against SARS-CoV-2 [14, 31] and a good correlation with virus neutralization (VN) titers. However, Salazar et al. in 2020 reported anti IgG-RBD to have slightly better correlation with VN in his study on 68 patients with COVID-19 [32]. In this data set, we were not able to directly test all samples here for virus neutralizing

activity. However, we have shown IgG to RBD as detected by the assay for neutralizing activity against SARS-CoV-2 and found this to correlation with IgG titers [33]. Therefore, we speculate that RBD antibodies found in our healthy blood donors reflected previous SARS-CoV-2 infection whilst a higher frequency of spike antibodies could be associated with cross-reactive antibodies. Cross-reactive antibodies against SARS-CoV-2 in pre-pandemic sera have been shown in studies from Africa [34, 35]. Moreover, a high prevalence of cross-reactive antibodies has been identified in sub-Saharan Africa [36]. While SARS-CoV-2 antibody positivity may protect against reinfection [37], the protective role of cross-reactive antibodies against SARS-CoV-2 needs further probing. No definitive conclusion can be drawn as we did not screen for antibodies against other coronaviruses in this study, due to lack of access to testing kits. Another limitation is that the study was conducted at a single institution and a relatively small sample size. Further, as with other seroprevalence studies, the rate of exposure may be under-estimated depending on the time of sampling after COVID-19 as antibody responses wane with time [38].

Factors such as, bacilli Calmette-Guerin vaccination [39], high temperature and humidity [40], and exposure to other viral/bacterial pathogens [41] have all been suggested in contributing to increasing protective immunity against SARS-CoV-2 in the population. High seroprevalence rates suggest active virus transmission in the community and highlights the importance of use of protective measures such as use of mask and social distancing.

Overall, this study provides important insights into SARS-CoV-2 seroprevalence rates in the local population just prior to the introduction of COVID-19 vaccination in February 2021. The high seroprevalence rates in this unvaccinated population reflect immunity which likely driven by exposure and enhanced by cross- reactive antibodies against SARS-CoV-2.

## Supporting information

**S1 Fig. Dilution curves in Elisa Assays IgG to COVID 19.** A positive serum pool containing high titers of anti convid19 antibodies was used to develop dilution curves. Serial 2-fold titrations from to 1/6400 were set up in each case. Graphs show an example of a dilution series for positive antibody pooled controls to either Spike protein (panel A) or RBD (panel B). A sigmoidal curve was obtained between a dilution of 50–3200 for both Spike and RBD proteins between OD 450 nm, 0.5–1.5.
(TIF)

**S2 Fig. Validation of upper and lower limits of the dilution curve in IgG ELISA to Spike and RBD proteins.** The variation between experiments for the higher and lower limits of dilution curves was determined for 10 consecutive assays for IgG antibodies to Spike (panel A), and RBD (panel B). The dotted lines indicate ± 2SD for each dilution. All other parameters are the same as in S1 Fig.
(TIF)

**S3 Fig. Monthwise seropositivity in blood donors in relation to their ABO blood groups.** The graphs show the number of individuals tested in December 2021 and February 2022 with different blood groups. IgG positive to spike (A) and to RBD (B) are depicted in each case. Standard deviation of percentage positivity is shows as error bars.
(TIF)

**S1 Table. Gender- and blood type-distribution in various age bands.**
(DOCX)

**S1 Appendix.**
(PDF)

## Acknowledgments

We thank IBET ITQB, Universidade NOVA de Lisboa, Portugal for assistance in providing recombinant proteins for ELISA assays. Thanks to Ambreen Wasim for statistical analysis. We thank the study subjects for participation in the study. We thank the Clinical Laboratory, AKUH, Karachi, Pakistan for facilitating sampling for the study.

## Author Contributions

**Conceptualization:** Muhammad Hasan, Bushra Moiz, Natasha Ali, J. Pedro Simas, Marc Veldhoen, Paula Alves, Syed Hani Abidi, Kulsoom Ghias, Erum Khan, Zahra Hasan.

**Data curation:** Muhammad Hasan, Shama Qaiser, Zara Ghous, Areeba Hussain.

**Formal analysis:** Muhammad Hasan, Shama Qaiser, Kiran Iqbal Masood, Zara Ghous, Areeba Hussain, Natasha Ali.

**Funding acquisition:** Zahra Hasan.

**Investigation:** Shama Qaiser, Zara Ghous, Areeba Hussain, Natasha Ali, Syed Hani Abidi, Erum Khan, Zahra Hasan.

**Methodology:** Shama Qaiser, Kiran Iqbal Masood, Zara Ghous, Areeba Hussain, Erum Khan, Zahra Hasan.

**Project administration:** Bushra Moiz, Kiran Iqbal Masood, Natasha Ali, Syed Hani Abidi, Kulsoom Ghias, Erum Khan, Zahra Hasan.

**Resources:** J. Pedro Simas, Marc Veldhoen, Paula Alves.

**Supervision:** Bushra Moiz, Kiran Iqbal Masood, Marc Veldhoen, Kulsoom Ghias, Erum Khan, Zahra Hasan.

**Validation:** Shama Qaiser, Kiran Iqbal Masood, Zara Ghous, Areeba Hussain, Zahra Hasan.

**Visualization:** Muhammad Hasan, Bushra Moiz, Kiran Iqbal Masood, Zara Ghous, Natasha Ali, J. Pedro Simas, Zahra Hasan.

**Writing – original draft:** Muhammad Hasan, Bushra Moiz, Kulsoom Ghias, Zahra Hasan.

**Writing – review & editing:** Bushra Moiz, Shama Qaiser, Kiran Iqbal Masood, Zara Ghous, Areeba Hussain, Natasha Ali, J. Pedro Simas, Marc Veldhoen, Paula Alves, Syed Hani Abidi, Kulsoom Ghias, Erum Khan, Zahra Hasan.

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
