## [Decision Letter · Decision Letter 0]

14 Mar 2022

PONE-D-21-30126Increasing IgG antibodies to SARS-CoV-2 in asymptomatic blood donors through the second COVID-19 wave in Karachi associated with exposure and immunity in the populationPLOS ONE

Dear Dr. Hasan,

Thank you for submitting your manuscript to PLOS ONE. After careful consideration, we feel that it has merit but does not fully meet PLOS ONE’s publication criteria as it currently stands. Therefore, we invite you to submit a revised version of the manuscript that addresses the points raised during the review process.

Two of the reviewers make a number of relevant constructive suggestions on how the manuscript can be improved. Please address all of these.==============================

We look forward to receiving your revised manuscript.

Kind regards,

Joël Mossong

Academic Editor

PLOS ONE

“The study was supported by a grant from Provost’s Academic Priorities Fund, The Aga Khan University, Pakistan”

Please state what role the funders took in the study.  If the funders had no role, please state: ""The funders had no role in study design, data collection and analysis, decision to publish, or preparation of the manuscript.

4. Thank you for stating the following in the Funding Section of your manuscript:

“The study was supported by a grant from Provost’s Academic Priorities Fund, The Aga Khan University.”

“The study was supported by a grant from Provost’s Academic Priorities Fund, The Aga Khan University, Pakistan”

5. Please remove your figures from within your manuscript file, leaving only the individual TIFF/EPS image files, uploaded separately.  These will be automatically included in the reviewers’ PDF.

Reviewers' comments:

Reviewer's Responses to Questions

**Comments to the Author**

1. Is the manuscript technically sound, and do the data support the conclusions?

Reviewer #1: Partly

Reviewer #2: Yes

Reviewer #3: Partly

2. Has the statistical analysis been performed appropriately and rigorously? 

Reviewer #1: No

Reviewer #2: Yes

Reviewer #3: N/A

3. Have the authors made all data underlying the findings in their manuscript fully available?

Reviewer #1: No

Reviewer #2: Yes

Reviewer #3: Yes

4. Is the manuscript presented in an intelligible fashion and written in standard English?

Reviewer #1: Yes

Reviewer #2: Yes

Reviewer #3: No

5. Review Comments to the Author

Reviewer #1: This is a review of "Increasing IgG antibodies to SARS-CoV-2 in asymptomatic blood donors through the second COVID-19 wave in Karachi associated with exposure and immunity in the population" from Hasan et al, 2022, for consideration at PLOS ONE.

This paper could use numerous clarifications prior to publication, which I (hopefully) constructively suggest below.

# Methods

1. Line 71 states: "Any donor with a history of COVID-19 related symptoms or confirmed infection was excluded from enrollment as a blood donor." HOWEVER, the whole premise of the paper is that you have found high seropositivity among these blood donors. And, in the discussion, you discuss the likely levels of asymptomatic infection which explain this confusion. I would suggest that, around Line 71, you explicitly note that this exclusion criterion in your methods does not prevent those who were asymptomatically or paucisymptomatically infected from donation, which is key to the present study.

2. Were the results adjusted for the sensitivity and specificity of the RBD and spike ELISAs? For that matter, what *were* the sensitivity and specificity of these assays? Were controls limited only to 100 known positive samples? Or were there also negative controls used? For instance, Figure 2 would be far better as seroprevalence estimates, with error bars, rather than raw sample counts which are not comparable between groups. Methods for adjustment are discussed in the context of SARS-CoV-2 here in https://elifesciences.org/articles/64206 and in general here https://www.hindawi.com/journals/eri/2011/608719/ In general, the establishment of the ELISA cutoffs should be more clearly discussed, with positive and negative controls noted or referenced.

# Results

2. On Line 145, I suggest noting for the reader that 99.1% of donors were male, so the result about 99.7% of spike+ being men is not significant. Same with the 29.0±7.4 and 29.2±7.2. A reader who is skimming the paper may be surprised by these numbers.

3. Do you have any estimate of how many of the samples might be RBD+ but spike-? On Line 294, the Discussion paragraph already notes that spike ELISAs are highly specific, so would you conclude that, perhaps, your RBD ELISA is relatively insensitive? If these numbers (RBD+, spike-) are not known, then this should be noted as a limitation.

4. "Our data showed an increasing frequency of seropositivity to spike and RBD in individuals who were between their second to fourth decades of life." Typically we would expect to see statistical tests of this claim, but I do not see them.

# Data Availability

Data were not provided alongside submission of the manuscript.

# Small Suggestions

- Perhaps the numbers on Line 3 (Abstract) could be updated since it is now February 2022.

- Refs 10 and 11 are the same paper, different title.

- L43-50 - place the literature review in dates or waves, but not "recent" or "previous". Years from now, it may be difficult to understand this paper without specific dates.

- BionaxNow spelling. (BinaxNow)

- A standard curve -[of] comprising a titration...

- Were tested for the presence of -[for] IgG antibodies... awkward.

- Fig 1 — what do the symbols and shading mean?

- "Similarly, though proportion of" -> "Similarly, the proportion of"

- "Overall, only 31.2% +[of] subjects having IgG to spike"

- Supplementary Fig 1 - the horizontal axis doesn't line up with the data points. What are the horizontal error bar type things? Please clarify what the colors represent, or annotate them on the plots.

Reviewer #2: The author can do a follow up spike and RBD antibodies in order to assess the concept of herd immunity in those positive cases. The work is good but is not novel as already one such study has been done in Karachi.

Reviewer #3: Authors attempted to assess SARS-CoV-2 seroprevalence in blood donors from Karachi, Pakistan, during the second COVID-19 pandemic wave.

There are several issues regarding this manuscript:

The small sample size, significant gender imbalance and the distribution of age groups. For instance they have 4 participants in the age group 51-60 years and the age range for group 1 is 17-20: 4 years. For an unbiased observation authors should have grouped participants into different age categories, for example: 17-25; 26-35; 36-45; 46-55.

It is not clear from the methods section the cut off for IgG positive test results. No information regarding the sensitivity and specificity of the test was provided.

The manuscript needs an English review. Some terms are inadequate and some sentences are hard to understand. Few examples:

Page 10: The current study performed December 2020 until February 2021 at the time

of the second COVID-19 wave shows a further rise (to 53%) in seropositivity in Karachi.

Page 11: the secondary questionnaire was administered later…

6. PLOS authors have the option to publish the peer review history of their article (what does this mean?). If published, this will include your full peer review and any attached files.

Reviewer #1: No

Reviewer #2: **Yes: **SAMRA WAHEED

Reviewer #3: No

---

## [Author Response · Author response to Decision Letter 0]

3 May 2022

PONE-D-21-30126

Increasing IgG antibodies to SARS-CoV-2 in asymptomatic blood donors through the pandemic in Karachi reflects exposure and immunity in the population

PLOS ONE

-Thank you. We have made the required style changes in the manuscript.

-Written consent from each study participant. Only adults aged 18 years and over were included in the study.

“The study was supported by a grant from Provost’s Academic Priorities Fund, The Aga Khan University, Pakistan”

Please state what role the funders took in the study. If the funders had no role, please state: ""The funders had no role in study design, data collection and analysis, decision to publish, or preparation of the manuscript.

-Thank you for pointing this out. This has been added to the cover letter. 

4. Thank you for stating the following in the Funding Section of your manuscript:

“The study was supported by a grant from Provost’s Academic Priorities Fund, The Aga Khan University.”

“The study was supported by a grant from Provost’s Academic Priorities Fund, The Aga Khan University, Pakistan”

- Thank you. This has been added to the cover letter.

5. Please remove your figures from within your manuscript file, leaving only the individual TIFF/EPS image files, uploaded separately. These will be automatically included in the -reviewers’ PDF.

-Thank you. We will upload the image files separately.

Reviewers' comments:

5. Review Comments to the Author

Reviewer #1: This is a review of "Increasing IgG antibodies to SARS-CoV-2 in asymptomatic blood donors through the second COVID-19 wave in Karachi associated with exposure and immunity in the population" from Hasan et al, 2022, for consideration at PLOS ONE.

This paper could use numerous clarifications prior to publication, which I (hopefully) constructively suggest below.

# Methods

1. Line 71 states: "Any donor with a history of COVID-19 related symptoms or confirmed infection was excluded from enrollment as a blood donor." HOWEVER, the whole premise of the paper is that you have found high seropositivity among these blood donors. And, in the discussion, you discuss the likely levels of asymptomatic infection which explain this confusion. I would suggest that, around Line 71, you explicitly note that this exclusion criterion in your methods does not prevent those who were asymptomatically or paucisymptomatically infected from donation, which is key to the present study.

- We thank the reviewer for this comment, the exclusion criteria has now been revised (line#80-81)

2. Were the results adjusted for the sensitivity and specificity of the RBD and spike ELISAs? For that matter, what *were* the sensitivity and specificity of these assays? Were controls limited only to 100 known positive samples? Or were there also negative controls used? For instance, Figure 2 would be far better as seroprevalence estimates, with error bars, rather than raw sample counts which are not comparable between groups. 

Methods for adjustment are discussed in the context of SARS-CoV-2 here in https://elifesciences.org/articles/64206 and in general here https://www.hindawi.com/journals/eri/2011/608719/ In general, the establishment of the ELISA cutoffs should be more clearly discussed, with positive and negative controls noted or referenced.

- thank you for pointing this out. We have now included the calculation for sensitivity and specificity of the RBD and spike ELISA assays in the manuscript. The assay validation is explained more clearly. We used 45 COVID-19 convalescent sera as positive controls and 55 healthy sera from pre-pandemic period as negative controls.

- We have added confidence intervals (95%) to the results to explain the range of antibody responses investigated in each condition tested.

# Results

2. On Line 145, I suggest noting for the reader that 99.1% of donors were male, so the result about 99.7% of spike+ being men is not significant. Same with the 29.0±7.4 and 29.2±7.2. A reader who is skimming the paper may be surprised by these numbers.

_ Thank you for pointing this out. We have re-written this line to avoid any confusion.

3. Do you have any estimate of how many of the samples might be RBD+ but spike-? On Line 294, the Discussion paragraph already notes that spike ELISAs are highly specific, so would you conclude that, perhaps, your RBD ELISA is relatively insensitive? If these numbers (RBD+, spike-) are not known, then this should be noted as a limitation.

- None of the samples were spike negative but RBD positive. We have expanded our discussion of the results to describe our observation. The sensitivity and specificity of spike and RBD ELISAs are discussed. IgG to both these proteins were significantly correlated with each other. 

- Further, it is noted that the dynamics of IgG to each of these proteins may differ in COVID-19.

4. "Our data showed an increasing frequency of seropositivity to spike and RBD in individuals who were between their second to fourth decades of life." Typically we would expect to see statistical tests of this claim, but I do not see them.

- Thank you for this suggestion. We have re-analysed the data according to the age groups suggested by the reviewer. After doing so, we did not find any difference between IgG seropositivity to spike or RBD across the age groups. This has been corrected in the manuscript.

# Data Availability

-Data availability statement is corrected. The data has been depicted in tables and figures used for the manuscript. The full set will be available upon acceptance of the publication. Additional material can be provided upon request.

# Small Suggestions

- Perhaps the numbers on Line 3 (Abstract) could be updated since it is now February 2022.

- Refs 10 and 11 are the same paper, different title.

- Thank you – this correction has been made.

- L43-50 - place the literature review in dates or waves, but not "recent" or "previous". Years from now, it may be difficult to understand this paper without specific dates.

- BionaxNow spelling. (BinaxNow)

 This correction has been made

- A standard curve -[of] comprising a titration...

Correction made

- Were tested for the presence of -[for] IgG antibodies... awkward.

correction made

- Fig 1 — what do the symbols and shading mean?

- "Similarly, though proportion of" -> "Similarly, the proportion of"

- "Overall, only 31.2% +[of] subjects having IgG to spike"

- Supplementary Fig 1 - the horizontal axis doesn't line up with the data points. What are the horizontal error bar type things? Please clarify what the colors represent, or annotate them on the plots.

- Thank you we have made this correction.

Reviewer #2: The author can do a follow up spike and RBD antibodies in order to assess the concept of herd immunity in those positive cases. The work is good but is not novel as already one such study has been done in Karachi.

-Thank you for this comment. We appreciate that there have been previous studies which have investigated the seroprevalence of antibodies to SARS-CoV-2. However, we believe it is of value to add to this body of literature in the context of different study populations and also locations across the country. Importantly, we discuss the value of using different antibody assays for identification of SARS-CoV-2 infections. Also, the possible significance of IgG to spike and RBD. Given that whilst spike is associated with protection to the virus, it is IgG to RBD which is directly correlated with neutralizing activity against SARS-CoV-2.

Reviewer #3: Authors attempted to assess SARS-CoV-2 seroprevalence in blood donors from Karachi, Pakistan, during the second COVID-19 pandemic wave.

There are several issues regarding this manuscript:

1. The small sample size, significant gender imbalance and the distribution of age groups. For instance they have 4 participants in the age group 51-60 years and the age range for group 1 is 17-20: 4 years. For an unbiased observation authors should have grouped participants into different age categories, for example: 17-25; 26-35; 36-45; 46-55.

-Thank you for this suggestion, we have re-analysed the data in the age categories as suggestion by the reviewer.

2. It is not clear from the methods section the cut off for IgG positive test results. No information regarding the sensitivity and specificity of the test was provided.

- We have added details regarding assay sensitivity and specificity and also cut-offs for IgG in the manuscript

3. The manuscript needs an English review. Some terms are inadequate and some sentences are hard to understand. Few examples:

- A review of the English in the manuscript as been conducted and corrections made. 

4. Page 10: The current study performed December 2020 until February 2021 at the time

of the second COVID-19 wave shows a further rise (to 53%) in seropositivity in Karachi.

- correction made

5. Page 11: the secondary questionnaire was administered later…

- correction made.________________________________________

---

## [Decision Letter · Decision Letter 1]

20 Jun 2022

PONE-D-21-30126R1Increasing IgG antibodies to SARS-CoV-2 in asymptomatic blood donors through the pandemic in Karachi reflects exposure and immunity in the populationPLOS ONE

Dear Dr. Hasan,

Thank you for submitting your manuscript to PLOS ONE. After careful consideration, we feel that it has merit but does not fully meet PLOS ONE’s publication criteria as it currently stands. Therefore, we invite you to submit a revised version of the manuscript that addresses the points raised during the review process.

ACADEMIC EDITOR:One of the reviewers has some minor comments. Please address these before resubmitting

We look forward to receiving your revised manuscript.

Kind regards,

Joël Mossong, PhD

Academic Editor

PLOS ONE

Journal Requirements:

Reviewers' comments:

Reviewer's Responses to Questions

**Comments to the Author**

1. If the authors have adequately addressed your comments raised in a previous round of review and you feel that this manuscript is now acceptable for publication, you may indicate that here to bypass the “Comments to the Author” section, enter your conflict of interest statement in the “Confidential to Editor” section, and submit your "Accept" recommendation.

Reviewer #1: All comments have been addressed

2. Is the manuscript technically sound, and do the data support the conclusions?

Reviewer #1: Yes

3. Has the statistical analysis been performed appropriately and rigorously? 

Reviewer #1: Yes

4. Have the authors made all data underlying the findings in their manuscript fully available?

Reviewer #1: No

5. Is the manuscript presented in an intelligible fashion and written in standard English?

Reviewer #1: Yes

6. Review Comments to the Author

Reviewer #1: # Review 2

I have only a few more suggestions, which are small in nature. Overall, the manuscript is improved, and I find that the authors have addressed my comments, as well as the comments of the other reviewer. I thank the authors for their attentiveness, and recommend publication after some small adjustments are made:

- My biggest suggestion is to consider a change in title. Between the two time points, the authors do not show increasing seroprevalence. This fact makes the title potentially misleading. One alternative title might be:

"IgG antibodies to SARS-CoV-2 in asymptomatic blood donors at two time points in Karachi"

or something straightforward and clear like that.

- I suggest stating in the first part of the abstract that individuals with known COVID-19 or symptoms in the prior 4 weeks were excluded.

- L168. "Hence" almost implies that the prior two percentages naturally lead to the third percentage. I suggest simply deleting "hence" to avoid this impression.

- The figures appear low-resolution and blurry to me. Please check before final submission.

7. PLOS authors have the option to publish the peer review history of their article (what does this mean?). If published, this will include your full peer review and any attached files.

Reviewer #1: No

---

## [Author Response · Author response to Decision Letter 1]

27 Jun 2022

Joël Mossong, PhD

Academic Editor

PLOS ONE

Dear Dr. Mossong,

Thank you for the Editorial and Reviewer feedback. We have addressed the concerns raised and hope that the manuscript will now be acceptable for publication in PLoSONE

Thank you

Best wishes,

Zahra Hasan

>>>

PONE-D-21-30126R1

Increasing IgG antibodies to SARS-CoV-2 in asymptomatic blood donors through the pandemic in Karachi reflects exposure and immunity in the population

PLOS ONE

Dear Dr. Hasan,

Thank you for submitting your manuscript to PLOS ONE. After careful consideration, we feel that it has merit but does not fully meet PLOS ONE’s publication criteria as it currently stands. Therefore, we invite you to submit a revised version of the manuscript that addresses the points raised during the review process.

ACADEMIC EDITOR:

One of the reviewers has some minor comments. Please address these before resubmitting

Reviewer 1

4. Have the authors made all data underlying the findings in their manuscript fully available?

Reviewer #1: No

>>

Response: We thank the reviewer for pointing this out. We have now included all supporting data used for the manuscript as S1 Appendix. 

6. Review Comments to the Author

Reviewer #1: # Review 2

I have only a few more suggestions, which are small in nature. Overall, the manuscript is improved, and I find that the authors have addressed my comments, as well as the comments of the other reviewer. I thank the authors for their attentiveness, and recommend publication after some small adjustments are made:

- My biggest suggestion is to consider a change in title. Between the two time points, the authors do not show increasing seroprevalence. This fact makes the title potentially misleading. One alternative title might be:

"IgG antibodies to SARS-CoV-2 in asymptomatic blood donors at two time points in Karachi"

or something straightforward and clear like that.

>>>

We thank the reviewer for their comments and have modified the title of the paper as suggested.

- I suggest stating in the first part of the abstract that individuals with known COVID-19 or symptoms in the prior 4 weeks were excluded.

>>

We thank the reviewer for their comments and have added this information in the Abstract

- L168. "Hence" almost implies that the prior two percentages naturally lead to the third percentage. I suggest simply deleting "hence" to avoid this impression.

>>

We have removed the work ‘Hence’ from L168

- The figures appear low-resolution and blurry to me. Please check before final submission.

>>

We thank the reviewer for their comments and have redone the TIF conversions for the files so that they are of better resolution

---

## [Editor Report · Decision Letter 2]

28 Jun 2022

IgG antibodies to SARS-CoV-2 in asymptomatic blood donors at two time points in Karachi

PONE-D-21-30126R2

Dear Dr. Hasan,

We’re pleased to inform you that your manuscript has been judged scientifically suitable for publication and will be formally accepted for publication once it meets all outstanding technical requirements.

Kind regards,

Joël Mossong, PhD

Academic Editor

PLOS ONE
---

## [Editor Report · Acceptance letter]

18 Jul 2022

PONE-D-21-30126R2 

IgG antibodies to SARS-CoV-2 in asymptomatic blood donors at two time points in Karachi 

Dear Dr. Hasan:

I'm pleased to inform you that your manuscript has been deemed suitable for publication in PLOS ONE. Congratulations! Your manuscript is now with our production department. 

Kind regards, 

on behalf of

Dr. Joël Mossong 

Academic Editor

PLOS ONE